# Diagnostic Accuracy and Interrater Agreement of FDG-PET/CT Lymph Node Staging in High-Risk Endometrial Cancer: The SENTIREC-Endo Study

**DOI:** 10.3390/cancers17142396

**Published:** 2025-07-19

**Authors:** Jorun Holm, André Henrique Dias, Oke Gerke, Annika Loft, Kirsten Bouchelouche, Mie Holm Vilstrup, Sarah Marie Bjørnholt, Sara Elisabeth Sponholtz, Kirsten Marie Jochumsen, Malene Grubbe Hildebrandt, Pernille Tine Jensen

**Affiliations:** 1Department of Nuclear Medicine, Odense University Hospital, 5000 Odense, Denmark; 2Department of Clinical Research, University of Southern Denmark, 5000 Odense, Denmark; 3Department of Nuclear Medicine and PET-Centre, Aarhus University Hospital, 8200 Aarhus, Denmark; 4Department of Clinical Physiology and Nuclear Medicine, Copenhagen University Hospital, 2200 Copenhagen, Denmark; 5Department of Clinical Medicine, Aarhus University, 8200 Aarhus, Denmark; 6Department of Radiology and Nuclear Medicine, Esbjerg and Grindsted Hospitals, 6700 Esbjerg, Denmark; 7Department of Gynecology and Obstetrics, Aarhus University Hospital, 8200 Aarhus, Denmark; 8Department of Gynecology and Obstetrics, Odense University Hospital, 5000 Odense, Denmark

**Keywords:** high-risk endometrial cancer, FDG-PET/CT, diagnostic accuracy, lymph node staging, interrater agreement

## Abstract

This study looked at how well FDG-PET/CT scans detect cancer spread to lymph nodes in women with high-risk endometrial cancer. It included 227 women undergoing treatment at three Danish hospitals. The scans correctly identified lymph node spread in many patients, especially along the major vessels, but were less reliable in finding all cases where cancer had spread to the lymph nodes. Specialists from different hospitals largely agreed on which lymph nodes looked suspicious, showing that the method was consistent across hospitals. While FDG-PET/CT alone cannot replace surgical staging, we confirmed that it added value when used alongside the sentinel lymph node mapping surgical procedure. The combined approach allows doctors to better identify the nodes that need removal for investigation, while avoiding unnecessary surgery with the removal of all lymph nodes. The findings support implementing FDG-PET/CT as part of a focused strategy of optimal lymph node staging in high-risk endometrial cancer.

## 1. Introduction

Lymph node metastasis is a significant negative prognostic factor in women with endometrial cancer (EC) [1,2]. Women with histologically confirmed lymph node metastases are allocated to adjuvant therapy because it improves survival [2,3,4]. The traditional surgical staging with radical pelvic and paraaortic lymph node dissection (PLD and PALD) in women with high-risk histology endometrial cancer (hrEC) has no benefit in itself but serves solely as a staging and treatment allocation procedure [5]. Radical lymph node dissection is associated with morbidities such as blood loss, nerve injury, and short- and long-term lymphoedema [6,7]. Less-invasive methods, such as the sentinel node technique, have been gradually implemented over the past decade [8,9,10]. The joint guidelines by the European Society of Gynaecological Oncology (ESGO), the European Society for Radiotherapy and Oncology (ESTRO) [11], and the European Society of Pathology (ESP) recommend sentinel lymph node (SLN) mapping in early-stage EC (low-risk and intermediate-risk disease) [12]. In high-intermediate and high-risk disease, surgical staging with lymphadenectomy has been recommended. Sentinel node mapping is increasingly accepted as an alternative to systematic dissection if performed using a sentinel node algorithm in the hands of experienced surgeons. The main limitation of SLN mapping is a potential failure to diagnose gross metastatic lymph node metastases that should be resected as part of the surgical treatment. Adding non-invasive imaging methods may mitigate this limitation and deserves further investigation.

In general, ^18^F-Fluorodeoxyglucose ([^18^F]FDG, henceforth FDG) positron emission tomography/computed tomography (PET/CT) is the most accurate imaging modality to detect lymph node metastases in gynaecological cancer, with high accuracy for lymph node diagnostics in advanced-stage endometrial, cervical, and vulval cancer [10,13,14]. In the latest 2023 classification by FIGO (Fédération Internationale de Gynécologie et d’Obstétrique), the staging of endometrial cancer relies on surgical–pathological findings along with molecular profiling, while imaging findings are not formally incorporated in the FIGO 2023 staging [9] or in the ESGO/ESTRO/ESP guidelines [12]. When evaluating FDG-PET/CT as a tool for lymph node diagnostics in women with endometrial cancer, the literature shows somewhat ambiguous results, and the accuracy is challenged, even though it outperforms both CT and magnetic resonance imaging [14,15,16,17,18,19,20].

The recent SENTIREC-endo high-risk study developed a new algorithm for sentinel node mapping in hrEC to replace the diagnostic radical dissection of lymph nodes [15]. This study concluded that safe clinical adoption would require the removal of any FDG-PET/CT-positive nodes for the diagnostic accuracy of the algorithm to be comparable to that of radical pelvic and paraaortic node dissection, thereby necessitating further evaluation of the role of PET/CT in the diagnostic workup by the gynaecological society.

The aim of this study was to investigate the diagnostic accuracy of FDG-PET/CT in the Danish SENTIREC-endo population to clarify how this imaging tool contributes to the staging of hrEC. The secondary objectives were to evaluate the importance of the location of lymph node metastases, especially in the paraaortic region, and to examine whether stratification by tumour size or the Standard Uptake Value (SUV) of the FDG uptake in the lymph nodes was useful as an evaluation tool. Finally, we aimed to validate the preoperative rating of the FDG-PET/CT by assessing interrater reliability between the three largest centres in Denmark specialised in the management of hrEC.

## 2. Materials and Methods

### 2.1. Participants and Design

This study on FDG-PET/CT in women with hrEC consisted of women who were prospectively included in a national Danish ‘SENTIREC’ multicentre database during the period from March 2017 to January 2023. The participating hospitals in Denmark were Copenhagen University Hospital, Aarhus University Hospital, and Odense University Hospital, which together manage 80% of women with high-risk EC in Denmark. As per protocol, all women were included with the intent to undergo an FDG-PET/CT scan as part of the preoperative diagnostic workup. The inclusion and exclusion criteria were the same as those used in the SENTIREC-endo high-risk study [21] and are given in Appendix A Appendix A. Table 1 shows the characteristics of the patients included in the present substudy. Figure 1 presents a flowchart of inclusion in the current FDG-PET/CT accuracy study. In the evaluation of the accuracy of FDG-PET/CT, we included all women in the SENTIREC-endo high-risk database who had a valid PET/CT performed and a valid reference standard pathology report following lymph node dissection. We did not exclude patients where the sentinel node procedure had failed or was incomplete, as long as we could obtain the pathology reference standard by systematic lymph node dissection. All participants provided written informed consent. The Danish Data Protection Agency (15/52037) and the Danish Independent Committees on Health Research (S-20150207) approved this study. We collected and managed the study data using the REDCap (Research Electronic Data Capture) tools version 13.7.18 hosted at Odense Explorative Network. The SENTIREC-endo trial was registered at ClinicalTrials.gov (NCT02820506).

### 2.2. Procedures

#### 2.2.1. Image Scanners and Procedures

Images were acquired according to international practice guidelines [22] with 3–4 MBq/kg body weight of FDG administered intravenously and the static PET scan performed 60 min after the injection of FDG. At Site 1 (Copenhagen) and Site 2 (Aarhus), the scans were primarily performed on a Siemens Biograph Vision 600 scanner and PET reconstructed using OSEM (Ordered Subset Expectation Maximisation) with time-of-flight (TOF) and a 2 mm Gaussian post-filter. At Site 3 (Odense), the scans were performed on a collection of GE Discovery MI (3, 4, and 5 rings) and GE Discovery 710 (3 rings) scanners with an OSEM reconstruction with TOF and a 5 mm Gaussian filter. At all sites, the CT scan was a diagnostic CT with intravenous iodinated contrast. At Sites 1 and 2, the CT scan was also supplemented with oral contrast. In general, the scanners were not certified by the EARL (European Association of Nuclear Medicine Research Ltd.) accreditation system. Local reconstructions were applied, and the imaging quality of these was generally superior to the minimum EARL standard.

#### 2.2.2. Image Interpretation and Comparison

Trained Nuclear Medicine and Radiology specialists assessed the FDG-PET/CT scans at all three centres. A clinical report of the imaging was prepared, and the evaluation of metastasis to the lymph nodes was recorded in a REDCap database using a standardised protocol. The first FDG-PET/CT rating registration was made before the surgical intervention to plan the procedure according to the FDG-positive lymph nodes and prior to obtaining the reference histology. A 5-point Likert scale was used by the imaging specialist to subjectively rate the suspicion of metastasis to the lymph nodes by FDG-PET/CT, ranging from 0 ‘no sign of metastasis’ to 1 ‘probably no sign of metastasis’, 2 ‘could be benign or malignant’, 3 ‘probably sign of metastasis’, and 4 ‘obvious sign of metastasis’. For overall accuracy, the FDG-PET/CT lymph node scores were dichotomised, where scores 0–1 were set to ‘test-negative’ and 2–4 were set to ‘test-positive’. All FDG-PET/CT-positive lymph nodes were registered in the database with their exact anatomical location. The surgical dissection was guided by the imaging for the removal and marking of the PET-positive nodes for registration in the database. This was performed in the same manner as with the marking and registration of nodes found by the sentinel node mapping and the removal of clinically suspicious nodes. This meant that one lymph node could be marked by one, two, or all three methods of detection. After marking the nodes, the radical surgical lymph node dissection in the pelvic and paraaortic regions was performed, thus obtaining the histology reference standard.

Regarding the detailed registration of the exact location of the PET-positive nodes, proximity to the aorta and/or the iliac vessels / the obturator fossa (left and right) with the estimated distance (mm) from fixed points like vessel bifurcations were described, allowing the surgeons to locate the PET-positive nodes precisely. The size of the lymph nodes by imaging were measured on the diagnostic CT. The SUVmax values of any positive lymph nodes were measured on the attenuated PET scan. The interrater agreement assessment of the FDG-PET/CT scans was performed postoperatively. The evaluations by each reader of the PET/CT scans were blinded to the other readers and to the reference standard results. In addition to the 5-point Likert scale (0–4), the ratings for interrater agreement were organised into three groups classified as ‘benign’ (Likert 0–1), ‘equivocal’ (Likert 2), and ‘malignant’ (Likert 3–4) for a more clinically appropriate estimation of agreement.

#### 2.2.3. Statistical Analyses

Power calculations were based on the negative predictive value for the sentinel node algorithm described in the main article [21]. We assessed the diagnostic accuracy by reporting overall accuracy (i.e., the proportion of true positive and true negative findings among all patients), sensitivity, specificity, positive predictive value, and negative predictive value. We supplemented these with 95% confidence intervals (95% CIs) based on the Wilson score method [23,24]. For area-under-the-receiver-operating-characteristic-curve (AUC-ROC) analyses, we reported point estimates and approximate Wald-type 95% CIs. Exploratory cut-off point determination was performed with Youden’s index [25].

Interrater agreement analyses comprised proportions of agreement [26] and Cohen’s Kappa [27], the latter supplemented by Wald-type 95% CIs. The level of significance was 5% (two-sided). We analysed the data with Stata/BE version 18 (StataCorp LP, College Station, TX 77845, USA).

## 3. Results

Of the 250 women initially enrolled in the SENTIREC-endo study, 227 patients were included in the FDG-PET/CT accuracy study; all had confirmed high-risk endometrial cancer, a valid preoperative PET/CT, and a valid reference standard histology report. Histology identified 52 women (23%) with lymph node metastasis. Eighteen women had lymph node metastases from EC in the paraaortic locations, described as being located above the inferior mesenteric artery. In six of these women, lymph node metastases were exclusively found in the paraaortic location. Surgical dissection of the paraaortic nodes was not performed in 20% of the women for clinically and surgically technical reasons, but pelvic lymphadenectomy was successfully completed for all women. A cross-tabulation of patients with PET-positive and PET-negative lymph nodes versus the findings by pathology reference standard is shown in Table 2.

### 3.1. Accuracy: By Patient and by Location

The overall accuracy for lymph node metastasis at the patient level identified by FDG-PET/CT was 83% [95% CI 77–87]. The sensitivity was 56% [95% CI 42–68] and the specificity was 91% [95% CI 86–94]. The positive predictive value (PPV) was 64% [95% CI 50–77] and the negative predictive value (NPV) was 87% [95% CI 82–91].

Based on the NPVs of PET, an illustration of the posterior probabilities of a negative test (1 – NPV) for lymph node metastasis related to the prior probability (prevalence of lymph node metastasis entering the study with hrEC = 52/227) is depicted in Figure 2. Figure 2 also presents the corresponding posterior probabilities following a stand-alone negative sentinel node mapping (9%), and the combination of the negative sentinel node algorithm with the negative FDG-PET/CT (2%) results from the algorithm study [21].

Stratified by paraaortic versus pelvic location, the sensitivity of FDG-PET/CT for metastasis to the paraaortic lymph nodes was similar to that of the pelvic lymph nodes: 56% [95% CI 34–75] versus 49% [95% CI 35–63]. The specificity of FDG-PET/CT for lymph node metastasis was higher in the paraaortic location compared to the pelvic locations: 97% [95% CI 93–98] versus 91% [95% CI 86–95]. The NPV was higher in the paraaortic location than in the pelvic area (97% vs. 87%), but the PPV was the same (59%).

### 3.2. Diagnostic Value of Lymph Node Size and FDG Standard Uptake Value (SUV)

The AUC-ROC curves for accuracy, assessed by lymph node sizes and SUVmax values in the FDG-avid lymph nodes, are presented in Figure A1 in the Appendix. For the lymph node size (largest size measured in mm), the AUC-ROC was 0.71 (95% CI 0.64–0.78), and the exploratory cut-point size was 5 mm (sensitivity/specificity = 0.51/0.91). For SUVmax, the AUC-ROC was 0.72 (95% CI 0.66–0.81), and the exploratory cut-point was SUV 3.95 (sensitivity/specificity = 0.47/0.97).

### 3.3. Interrater Agreement

Interrater reliability was assessed to evaluate the diagnostic agreement between the three participating Nuclear Medicine departments from each hospital centre. Patient-based absolute agreement, corresponding Cohen’s Kappa values, and 95% confidence intervals are reported in Table 3. Due to delayed second readings performed after the women had their operation, four scans from Odense University Hospital (Site 3) lacked the necessary imaging data and were excluded from this analysis (total *n* = 223). Overall, the absolute agreement was 95% with a Cohen’s Kappa value of *κ* = 0.84 (95% CI: 0.73–0.94), indicating ‘almost perfect’ agreement. Agreement was also stratified by which centre the scans originated from. At Site 3, three different raters contributed to the initial readings due to clinical scheduling. The highest agreement (*κ* = 1.0) was observed for scans performed in Aarhus (Site 2), evaluated by raters from Aarhus and Odense. In the paraaortic locations, the overall interrater agreement was higher than for the pelvic locations, with an absolute agreement of 0.96 and a Kappa value of 0.73 (agreement strictly location-based and not patient-based). See Table 4.

## 4. Discussion

To our knowledge, the SENTIREC-endo study is the largest prospective investigation evaluating the diagnostic performance of the sentinel node algorithm in combination with the accuracy of FDG-PET/CT in hrEC using an optimal reference standard that included systematic radical pelvic and, in most cases, paraaortic lymphadenectomy. The 23% prevalence of lymph node metastasis we observed in hrEC in our study aligns with the reported rates in the literature (17–23%) [28,29]. In contrast, lymph node metastasis in low-risk endometrial cancer is as rare as 0.3% [30]. Therefore, we considered it relevant to investigate the role of FDG-PET/CT as a less-invasive diagnostic tool for lymph node assessment, specifically in hrEC.

The diagnostic performance of FDG-PET/CT for staging lymph node metastasis in this study demonstrates limited sensitivity but high specificity and negative predictive value (NPV), which aligns with previous findings [31]. Due to its limited sensitivity, FDG-PET/CT cannot replace surgical lymph node dissection as a stand-alone diagnostic modality. However, as a non-invasive and less harmful tool than radical dissection, its potential role in preoperative staging remains of clinical importance. From a clinical and patient-centred perspective, the most relevant question is the individual probability of having lymph node metastasis, particularly following a negative test, where unnecessary treatment is generally avoided. Figure 2 illustrates the change in the estimated prior probability for a woman with high-risk endometrial cancer to have lymph node metastasis, which is represented by the 23% prevalence of lymph node metastasis prior to the diagnostic tests (52/227 women). In cases where the preoperative FDG-PET/CT yields a negative result, indicating no signs of lymph node metastasis, the posterior probability (1–NPV) following negative FDG-PET/CT drops to 13%. This level of certainty is insufficient to reliably exclude metastasis, highlighting the necessity of additional diagnostic evaluation, such as SLN assessment, to ensure accurate staging and guide treatment decisions.

Figure 2 further illustrates that the combination of FDG-PET/CT and sentinel node mapping approximates the diagnostic accuracy of full lymph node dissection—yielding a specificity of 95% and an NPV of 98% [21], which corresponds to a 2% posterior probability of lymph node metastasis if the joint methods are all negative. The intuitive question is why the combined NPV of PET staging and SLN mapping adds up to 98% instead of approaching a value of around 90%, like the NPV of the two isolated methods. However, the two methods mainly reflect the findings in two different lymph node locations: the sentinel node for lymph node metastasis in the pelvis and the FDG-PET/CT in the paraaortic location, resulting in an additive effect for joint accuracy. For lymph node metastases in the paraaortic location, the FDG-PET/CT has an NPV of 97% compared to the overall NPV of 87%. In the SENTIREC-endo study, sentinel-node-positive nodes in the paraaortic nodes occurred in only 0.8% of patients and only below the inferior mesenteric artery. Routine lymph node dissection above the inferior mesenteric artery increases surgical time and the risk of postoperative morbidity [6]. Even for experienced surgeons, the procedure can be challenging and is often performed at the surgeon’s discretion. Incorporating FDG-PET/CT, particularly for identifying PET-positive paraaortic nodes cranial to this level, significantly improved the combined staging accuracy. The high NPV of FDG-PET/CT for paraaortic metastases (97%) suggests that PET-negative paraaortic nodes are most likely not malignant, and the posterior probability of metastasis in these locations with negative FDG-PET/CT alone is only 3%. This permits a more individualised approach, allowing surgeons to decide whether to perform paraaortic lymph node dissection when FDG-PET/CT findings are negative in those regions. However, this demands a bilateral pelvic SNL mapping procedure according to the sentinel node algorithm, as previously described in the SENTIREC-endo study [21].

Notably, none of the paraaortic metastases above the inferior mesenteric artery were detected by the sentinel node mapping alone, thus highlighting the added value of FDG-PET/CT for comprehensive paraaortic staging. Sentinel node staging does not allow for the calculation of specificity, as it inherently produces no false positives. However, the rate of false positives is less critical in this context; what matters most for clinical utility is the ability to reliably rule out nodal metastases, where both sensitivity and NPV are key. From this perspective, less-invasive approaches are preferable to radical dissection, provided they offer comparable reassurance.

Earlier accuracy studies for lymph node diagnostics in hrEC are mostly retrospective, with varying results [20]. Fasmer et al. reported a sensitivity and specificity of 56% and 90%, respectively, though only 39% in their cohort had high-risk EC [15]. In another single-centre study, Nordskar et al. reported 63% sensitivity and 98% specificity, with 48% of PET-positive patients classified as high-risk [12]. A recent prospective study reported 46% sensitivity and 91% specificity for extrauterine dissemination that included both lymph node and peritoneal dissemination, comparable to our 56% sensitivity and similarly high specificity [21]. However, paraaortic dissection in that study was inconsistently applied and not well documented, weakening the histological reference standard.

Focusing on paraaortic staging, a retrospective French multicentre study in high-risk EC found slightly higher sensitivity (61.8%) but lower specificity (89.7%) [14]. Nordskar et al. reported 100% sensitivity for isolated paraaortic nodes [12], likely inflated by selective dissection based on preoperative imaging and performed solely at the surgeon’s discretion. This methodological bias—where paraaortic lymphadenectomy was triggered by imaging findings—likely overestimates sensitivity by excluding histologically undetected metastases. In contrast, the SENTIREC-endo study pursued the systematic dissection of lymph nodes up to the renal vein in 80% of cases, providing a more reliable reference standard. This likely explains our lower, but more accurate, sensitivity estimates for the paraaortic nodes.

Despite a high specificity for paraaortic nodes (97%), the PPV remained moderate due to six false positives. In one instance, histology revealed lymphoma, which the PET assessors had correctly suspected in their report. However, the nodes had to be scored as Likert 3 (‘probable sign of metastasis’) and were thus categorised as PET-positive for EC metastasis. With this in mind, a PPV—or posterior probability of a positive PET/CT—of 59% for paraaortic nodes underscores the clinical necessity of resecting PET-positive nodes. FDG-avid nodes almost always indicate an underlying pathology, even if not attributable to endometrial carcinoma.

### Limitations

A randomised controlled trial would have been the optimal format for evaluating a clinical diagnostic strategy. Such a design would have enabled the direct comparison of the SNL + FDG-PET/CT algorithm with alternative strategies. Another limitation is that patient inclusion occurred prior to the 2023 FIGO classification update, which now incorporates molecular profiling [9]. Future studies should integrate molecular classification into risk stratification to further refine diagnostic algorithms in endometrial cancer.

Additionally, the study was conducted in Denmark, a small and relatively homogenous country with universal, publicly funded healthcare. While this setting reduces the risk of confounding due to socioeconomic disparities, it may limit the generalisability of our findings to more ethnically and culturally diverse populations. These factors should be carefully considered in the design of future clinical trials focused on lymph node diagnostics in high-risk endometrial cancer. The value of quantitative PET metrics, such as SUVmax or lymph node size, proved limited. The ROC curves (Figure A1) were skewed because only FDG-avid and visible nodes could be measured, excluding the substantial proportion of nodes scored as ‘Likert 0′. A further limitation was that SUV measurements were based on local PET OSEM reconstructions rather than EARL-compliant reconstructions, thereby restricting the comparability of SUVs across clinical sites [32].

Neither lymph node size nor SUVmax produced clinically meaningful cut-off values (AUC ~0.7), suggesting the limited utility of semi-quantitative thresholds. Future developments may instead depend on machine-learning-based image analysis rather than fixed quantitative metrics.

In the absence of robust quantitative tools, interrater reliability becomes crucial. With participation from the three major Danish hospital centres, our study demonstrates that the FDG-PET/CT interpretation was consistent nationwide, achieving 95% agreement and a Cohen’s Kappa of 0.84—classified as ‘almost perfect’. Agreement was even higher for paraaortic nodes (96%) compared to pelvic nodes, highlighting the reliability of PET interpretation in this anatomically complex region and reinforcing the importance of surgical verification when paraaortic nodes appear suspicious.

## 5. Conclusions

In this large, prospective, and multicentre study, FDG-PET/CT demonstrated moderate sensitivity but high specificity in the staging of lymph node metastases among women with high-risk endometrial cancer. While FDG-PET/CT alone is not sufficient to replace surgical staging, its high specificity and excellent interrater agreement across centres underscore its value in guiding the targeted dissection of PET-positive lymph nodes. The SENTIREC-endo algorithm, which integrates sentinel node mapping with PET-directed nodal excision, offers a safe and effective alternative to radical pelvic and paraaortic lymphadenectomy for accurate lymph node staging.

## Figures and Tables

**Figure 1 cancers-17-02396-f001:**
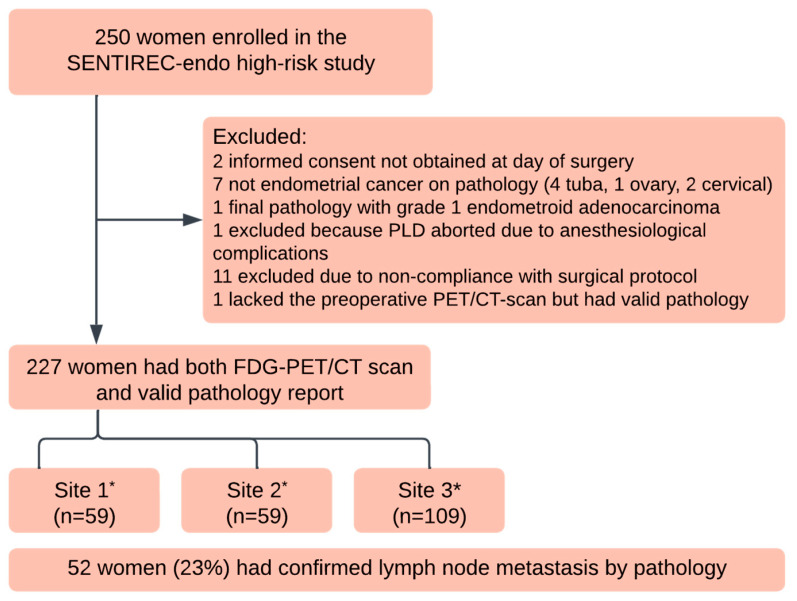
Flowchart of the SENTIREC-endo high-risk FDG-PET/CT accuracy and interrater substudy. Site 1*: Copenhagen University Hospital, Site 2*: Aarhus University Hospital, Site 3*: Odense University Hospital.

**Figure 2 cancers-17-02396-f002:**
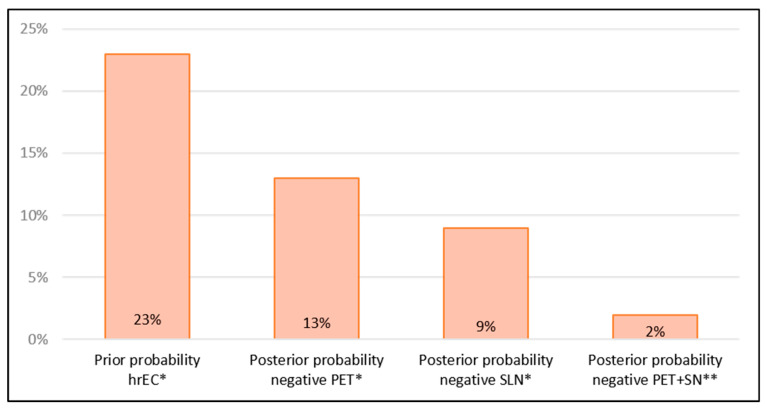
Prior probability vs. posterior probabilities of a negative test for lymph node metastasis for PET*, SLN*, and the combined PET + SLN** algorithm. hrEC* = high-risk endometrial cancer, PET represents FDG-PET/CT, SNL represents sentinel node mapping. PET + SLN** represents the PET + SNL algorithm that includes pelvic and paraaortic lymphadenectomy in case of failed SLN.

**Table 1 cancers-17-02396-t001:** Demographic and histological characteristics of women with high-risk endometrial cancer.

Women with hrEC Included in FDG-PET/CT Substudy (*n* = 227)	No.	(Range or %)
Age (mean)	70	44–88
BMI ^a^ (mean)	27.9	17.0–46.7
**Lymph node dissection**		
SLN ^b^+ PLD ^c^ only	45	19.8
SLN ^b^ + PLD ^c^+ PALD ^d^	165	72.7
SLN ^b^ + PLD ^c^ + removal of selected paraaortic nodes	5	2.2
PLD ^c^ + PALD ^d^ (SLN ^b^ not performed according to surgery algorithm)	12	5.3
**Final FIGO stage**		
IA	125	55.0
IB	21	9.3
II	18	8.0
IIIA	5	2.2
IIIB	4	1.7
IIIC1	34	15.0
IIIC2	12	5.3
IVA	0	0.0
IVB	8	3.5
**Final histology** (more than one could be applied)		
Endometroid adenocarcinoma grade III	60	26.4
Serous adenocarcinoma	109	48.0
Clear cell adenocarcinoma	40	17.6
Un- or dedifferentiated carcinoma	4	1.7
Carcinosarcoma	36	15.9
Mixed type II histology	3	1,3
Mesonephric-like adenocarcinoma	1	0.4
**Location of lymph node involvement by FDG-PET/CT**		
Pelvic lymph node involvement	35	15.4
Paraaortic lymph node involvement	18	7.9
Isolated pelvic involvement	27	11.9
Isolated paraaortic involvement	6	2.6
**Lymphovascular space invasion**	44	19

^a^ Body Mass Index; ^b^ SLN = sentinel node procedure; ^c^ PLD = pelvic lymph node dissection; ^d^ PALD = paraaortic lymph node dissection.

**Table 2 cancers-17-02396-t002:** Cross-tabulation of PET-positive and PET-negative metastatic lymph node results versus pathology-verified results in women with high-risk endometrial cancer.

FDG-PET/CT vs. Pathology Report as Reference Standard	Number of Women with Negative Lymph Nodes Determined by Pathology Report	Number of Women with Positive Lymph Nodes Determined by Pathology Report	Total
Number of women with negative lymph nodes on FDG-PET/CT	159	23	182
Number of women with positive lymph nodes on FDG-PET/CT	16	29	45
**Total number of women with lymph node dissection performed**	175	52	227

**Table 3 cancers-17-02396-t003:** Interrater agreement between the three Danish hospitals: overall agreement and agreement between hospitals.

Sites of Patients	Sites of Raters	Absolute Agreement	Cohen’s Kappa	95% CI for Kappa
All sites (*n* = 223) *	Sites 1, 2, and 3	0.95	0.84	0.73–0.94
Site 1 (*n* = 59)	Site 1 vs. Site 3	0.97	0.91	0.67–1
Site 2 (*n* = 59)	Site 2 vs. Site 3	1	1	N/A
Site 3* (*n* = 105)	Site 3 vs. Site 2	0.92	0.62	0.47–0.77

* Four of the scans originating from Site 3 did not have complete data for the second interrater evaluation

**Table 4 cancers-17-02396-t004:** Interrater agreement by lymph node location for all three hospital units: Copenhagen, Aarhus, and Odense.

Location of FDG-Positive Lymph Nodes	Absolute Agreement	Cohen’s Kappa	95% CI for Kappa
Pelvic location	0.89	0.65	0.52–0.78
Paraaortic location	0.96	0.73	0.60–0.86

## Data Availability

The raw data of the SENTIREC-endo study is unavailable for publication due to privacy or ethical restrictions. It is hosted at the Odense Explorative Network with restricted availability only to the research project managers by regulation of the Danish Data Protection Agency (protocol code: 15/52037, approval date: 15 June 2016).

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
