# Peer review of "Diagnostic Accuracy and Interrater Agreement of FDG-PET/CT Lymph Node Staging in High-Risk Endometrial Cancer: The SENTIREC-Endo Study"

_cancers, 2025, doi:10.3390/cancers17142396_

Round 1

Reviewer 1 Report

Comments and Suggestions for Authors

This study provides significant clinical evidence for lymph node staging in high-risk endometrial cancer, supporting the feasibility of combining FDG-PET/CT with sentinel lymph node (SLN) mapping. Despite certain limitations, the research is rigorously designed, yields reliable results, and holds substantial academic and clinical value.In general, the work of this paper is clear and logical, but some points are required to be modified.

1.This research lacks long-term prognostic data. Future studies should analyze how the combined approach affects survival outcomes.

2.The study was exclusively based on Danish patients. So the authors could consider including participants from more diverse countries and ethnic backgrounds to validate the generalizability of the findings in future research.

Author Response

Thank you very much for taking the time to review our manuscript. Please find the detailed responses below and the corresponding revisions/corrections in track changes in the re-submitted files.

REVIEWER 1

This study provides significant clinical evidence for lymph node staging in high-risk endometrial cancer, supporting the feasibility of combining FDG-PET/CT with sentinel lymph node (SLN) mapping. Despite certain limitations, the research is rigorously designed, yields reliable results, and holds substantial academic and clinical value. In general, the work of this paper is clear and logical, but some points are required to be modified.

Comment 1: This research lacks long-term prognostic data. Future studies should analyze how the combined approach affects survival outcomes.

Response 1: We fully agree with the reviewer’s point. Long-term prognostic data are indeed crucial to assess the clinical impact of the combined diagnostic approach. We are currently collecting and finalizing survival and outcome data on the 170 patients included in the SENTIREC-endo algorithm study who underwent both the appropriate sentinel lymph node procedure and FDG-PET/CT. These data will form the basis of a future manuscript focused specifically on survival outcomes.

Comment 2. The study was exclusively based on Danish patients. So the authors could consider including participants from more diverse countries and ethnic backgrounds to validate the generalizability of the findings in future research.

Response 2: Thank you for this important comment. We fully agree. Although this national study includes patients from across the Kingdom of Denmark and, to our knowledge, represents the largest prospective investigation of lymph node diagnostics in women with high-risk endometrial cancer, it remains a limitation that the cohort is restricted to a relatively small and demographically homogenous population. Denmark’s universal, publicly funded healthcare system ensures equal access to care, which may not reflect conditions in more diverse or resource-variable settings. While inclusion in the SENTIREC study is now closed, future research should indeed prioritize broader international and ethnically diverse populations to better validate the generalizability of these findings. We have addressed this limitation, along with others, in a newly added section (4.1 Limitations) in the discussion, page 10, lines 353-358.

Reviewer 2 Report

Comments and Suggestions for Authors

This large, prospective multicenter study evaluates the diagnostic performance and interrater agreement of FDG-PET/CT for lymph node staging in high-risk endometrial cancer. The study supports a combined strategy of FDG-PET/CT and sentinel lymph node mapping as a less invasive yet accurate alternative to full pelvic and paraaortic lymphadenectomy.

Although the manuscript can be considered already of high quality, I would suggest taking into account the following recommendations:

Minor grammatical adjustments and rephrasing for conciseness would improve readability.

Consider emphasizing the clinical impact of the combined PET/CT + SLN strategy more clearly in the abstract conclusion.

The intro could benefit from clarification that imaging is not formally incorporated into FIGO 2023 staging.

Consider adding a PRISMA-style flow diagram in the supplementary materials for full transparency on inclusion/exclusion.

Author Response

Thank you very much for taking the time to review our manuscript. Please find the detailed responses below and the corresponding revisions/corrections in track changes in the re-submitted files.

This large, prospective multicenter study evaluates the diagnostic performance and interrater agreement of FDG-PET/CT for lymph node staging in high-risk endometrial cancer. The study supports a combined strategy of FDG-PET/CT and sentinel lymph node mapping as a less invasive yet accurate alternative to full pelvic and paraaortic lymphadenectomy. Although the manuscript can be considered already of high quality, I would suggest taking into account the following recommendations:

Comment 1: Minor grammatical adjustments and rephrasing for conciseness would improve readability.

Response 1: The manuscript has undergone a general revision to improve grammar and language in British English.

Comment 2: Consider emphasizing the clinical impact of the combined PET/CT + SLN strategy more clearly in the abstract conclusion.

Response 2: Thank you for this valuable suggestion. We have added the word “clinical” to the Abstract Conclusion to better emphasize the clinical implementation of the combined PET/CT and sentinel lymph node (SLN) procedure for targeted lymph node dissection (abstract, page 2, line 47).

The combined sentinel node and FDG-PET/CT algorithm results were previously published in the SENTIREC-endo algorithm article by Bjørnholt et al. The purpose of this current manuscript, following feedback from prior peer reviewers, is to clarify the specific role of FDG-PET/CT lymph node diagnostics within the SENTIREC-endo study—both in comparison to the sentinel node procedure alone, the combined diagnostic algorithm, and the radical lymph node dissection reference standard.

Therefore, this study focuses on concluding the diagnostic accuracy of the modalities, particularly FDG-PET/CT. We strongly believe that if the gynaecological community moves toward replacing radical lymph node dissection with a less invasive yet accurate sentinel node algorithm for high-risk endometrial cancer staging, this study supports including FDG-PET/CT in preoperative planning to ensure safe and precise targeted lymph node dissection.

Comment 3: The intro could benefit from clarification that imaging is not formally incorporated into FIGO 2023 staging.

Response 3: Thank you for this helpful suggestion. We have incorporated your recommended wording into the manuscript, specifically in the ‘1. Introduction’ section (page 2, lines 79–80), to clarify that imaging is not formally included in the FIGO 2023 staging system.

Comment 4: Consider adding a PRISMA-style flow diagram in the supplementary materials for full transparency on inclusion/exclusion.

Response 4: Thank you for this valuable suggestion. As access to the inclusion and exclusion criteria was also requested by another reviewer, we have added a detailed table outlining these criteria in the Supplementary Materials (Table S1). A reference to this table has been included in the ‘2. Materials and Methods’ section (page 3, lines 107–108). A flowchart of the further selection for this sub-study on the role of FDG-PET/CT is depicted in Figure 1 in the revised manuscript.

Reviewer 3 Report

Comments and Suggestions for Authors

This is a well-conducted and well-written study. I have mostly minor suggestions to improve its clarity.

  1. It is not very convenient to address another published paper to learn about the inclusion/exclusion criteria. It would be better if the authors list them briefly in the "Participants and design" subsection. Alternatively, they can be listed in a supplementary file.
  2. The same comment relates to the sentinel node algorithm. The reader is supposed to read another manuscript to better understand the methodology of the current one.
  3. Also, the discussion of study limitations need to be included.

Otherwise, this study helps is based on a large cohort and is very robust. I believe that it deserves publication in the Cancers after those minor comments are addressed.

Author Response

Thank you very much for taking the time to review our manuscript. Please find the detailed responses below and the corresponding revisions/corrections in track changes in the re-submitted files.

This is a well-conducted and well-written study. I have mostly minor suggestions to improve its clarity.

Comment 1: It is not very convenient to address another published paper to learn about the inclusion/exclusion criteria. It would be better if the authors list them briefly in the "Participants and design" subsection. Alternatively, they can be listed in a supplementary file.

Response 1: Thank you for this helpful suggestion. Since access to the inclusion and exclusion criteria was also requested by another reviewer, we have now included a detailed list of these criteria in the Supplementary Materials (Table S1). A reference to this table has been added in the ‘Materials and Methods’ section (page 3, lines 107–108) for easier access and transparency.

Comment 2: The same comment relates to the sentinel node algorithm. The reader is supposed to read another manuscript to better understand the methodology of the current one.

Response 2: Thank you for this very relevant comment. We understand that having to refer to another article can be inconvenient for readers. To address this, the SENTIREC-endo surgical procedure — covering the registration of sentinel, PET-positive, and clinically suspicious lymph nodes — is briefly described in the Materials and Methods section (page 5, lines 157–164).

This sub-study primarily focuses on the PET/CT imaging methodology, including scan interpretation and comparative analyses. Following requests from peer reviewers of the original algorithm article, our goal here was to clarify the individual and combined diagnostic contributions of PET/CT and sentinel node procedures in a real-world clinical setting.

While it would have been possible to focus solely on direct comparisons between diagnostic modalities, we believe this approach offers less clinical relevance. Previous retrospective studies have done this with variable accuracy. Currently, radical lymph node dissection remains the gold standard for lymph node staging in high-risk endometrial cancer despite advances in sentinel node techniques. However, due to the invasive and morbid nature of radical dissection, exploring combined, less invasive diagnostic strategies—such as PET/CT plus sentinel node mapping—is essential. This sub-study therefore complements the previously published SENTIREC-endo algorithm results by specifically elucidating the role of PET/CT in this combined diagnostic pathway.

Comment 3: Also, the discussion of study limitations need to be included.

Response 3: Thank you for highlighting this important point. We initially followed the Cancers journal’s article template, which does not specifically include a separate “Limitations” section. Consequently, our discussion of study limitations was integrated within the broader ‘4. Discussion’ section. We recognize that this may have made the limitations less accessible to readers.

To improve clarity, we have now added a dedicated subsection titled ‘4.1 Limitations’ within the Discussion. This new section consolidates the relevant limitations, including additional considerations raised by all peer reviewers, to ensure these critical points are more clearly presented and easier to locate.

Otherwise, this study helps is based on a large cohort and is very robust. I believe that it deserves publication in the Cancers after those minor comments are addressed.

Reviewer 4 Report

Comments and Suggestions for Authors

Dear Authors,
I read your article with much interest. The possibility of improving the diagnostic accuracy of non-invasive methods in order to reduce surgical risks is very important. In endometrial cancer, removal of lymph nodes is not curative but stadiative, so reducing lymphanedectomy is essential.
The main limitation of the study is the lack of a control group in which CT-Total Body is routinely done to study endometrial cancer at a distance and thus evaluate the advantages of PET-CT over CT alone.
However to improve the article I suggest:
-discuss the diagnostic accuracy advantage of PET-CT over CT alone compared to literature data
-discuss the possible application of PET-CT only in cases of endometrial cancer of high histological grade or high biomolecular risk
-underline the advantages in terms of reduction of upper abdominal dissection in the presence of negative PET-CT for para-aortic lesions
-underline in the limitations of the study the absence of a comparison group with only CT

Best Regards

Author Response

Thank you very much for taking the time to review our manuscript. Please find the detailed responses below and the corresponding revisions/corrections in track changes in the re-submitted files.

Dear Authors,
I read your article with much interest. The possibility of improving the diagnostic accuracy of non-invasive methods in order to reduce surgical risks is very important. In endometrial cancer, removal of lymph nodes is not curative but stadiative, so reducing lymphadenectomy is essential.
The main limitation of the study is the lack of a control group in which CT-Total Body is routinely done to study endometrial cancer at a distance and thus evaluate the advantages of PET-CT over CT alone.
However to improve the article I suggest:

Comment 1: Discuss the diagnostic accuracy advantage of PET-CT over CT alone compared to literature data.

Response 1: Thank you for this suggestion. We have in the Introduction section added a comment on the advantages of PET/CT over CT, as well as its comparative advantages over MRI for lymph node diagnostics, with appropriate references to the literature. These additions can be found in the ‘1. Introduction’ section, page 2, line 83.

Comment 2: Discuss the possible application of PET-CT only in cases of endometrial cancer of high histological grade or high biomolecular risk.

Response 2: Thank you for this thoughtful suggestion. We have added a paragraph in the ‘4. Discussion’ section (page 8, lines 261–266) comparing the prevalence of lymph node metastasis in high-risk versus low-risk endometrial cancer. This addition supports our decision to focus exclusively on high-risk patients in this study and clarifies the rationale for incorporating FDG-PET/CT into the diagnostic workup and surgical algorithm specifically for this subgroup.

Comment 3: Underline the advantages in terms of reduction of upper abdominal dissection in the presence of negative PET-CT for para-aortic lesions.

Response 3: Thank you for this very helpful suggestion. Reduction of upper abdominal dissection is indeed what we aim for, and this should be underscored in the manuscript. To clarify the advantage of a negative PET/CT for paraaortic lymph node assessment, we have revised and reorganized the relevant part of the ‘4. Discussion’ section (to page 9, lines 299–302), and added new supporting text in lines 294–297 and 302–306 to highlight the potential for individual assessment of not performing upper abdominal dissection when PET/CT findings are negative.

Comment 4: Underline in the limitations of the study the absence of a comparison group with only CT.

Response 4: Thank you for this valuable suggestion. We have addressed this point in the newly added subsection ‘4.1 Limitations’ of the ‘4. Discussion’ section (see page 10, lines 347–352), where we note that a randomized study design, including a comparison group, would have been optimal. We also include additional limitations considerations inspired by reviewer’s comments regarding planning of future studies that should include the molecular profiling risks that are now included in the FIGO 2023 classification. Also considerations regarding the generalizability to international and ethnically diverse populations in the planning of future studies is added (page 10, lines 353–358).

In Denmark, FDG-PET/CT has been the routine imaging modality for patients with high-risk endometrial cancer or suspected metastases for over 15 years, because it is the better imaging modality for diagnosing metastasis also outside the lymph nodes, and has been part of the Danish National Guidelines preoperative diagnostic work-up for many years. CT alone is generally not performed unless a patient cannot undergo PET/CT—an uncommon situation. We acknowledge that PET/CT is not widely accessible in many parts of the world, and it would be highly relevant to assess the accuracy of a sentinel node algorithm combined with CT alone, ideally in a randomized setting.

In the SENTIREC project, sentinel node procedures were guided by PET/CT findings, so we cannot retrospectively isolate the CT component for a fully prospective analysis. However, as you rightly suggest, a future sub-study could involve a blinded re-evaluation of the CT portion of the PET/CT scans—using only the referral text and excluding other clinical data. This could indeed form a valuable dataset for comparison, and we appreciate you drawing our attention to this possibility.

Round 2

Reviewer 4 Report

Comments and Suggestions for Authors

Dear authors,
the revised manuscript has improved. Some limitations remain, which you explained in the discussion.

Best regards